# Effect of Date Palm Pollen Supplementation on the Egg Production, Ovarian Follicles Development, Hematological Variables and Hormonal Profile of Laying Hens

**DOI:** 10.3390/ani11010069

**Published:** 2021-01-01

**Authors:** Mohamed Saleh, Dariusz Kokoszyński, Mohamed Abd-Allah Mousa, Ahmed Abdel-Kareem Abuoghaba

**Affiliations:** 1Department of Poultry Production, Faculty of Agriculture, Sohag University, Sohag 82524, Egypt; mohamed.saleh@agr.sohag.edu.eg (M.S.); Abuoghaba@yahoo.com (A.A.-K.A.); 2Department of Animal Sciences, Faculty of Animal Breeding and Biology, UTP University of Science and Technology, 85084 Bydgoszcz, Poland; 3Department of Nutrition and Clinical Nutrition, Faculty of Veterinary Medicine, Sohag University, Sohag 82524, Egypt; Dr_m_mousa@yahoo.com

**Keywords:** DPP, laying hens, egg quality, ovarian functions, hormonal profile

## Abstract

**Simple Summary:**

Date palm pollen (DPP) is a natural product produced from male palm flowers and is used to improve ovulation and fertility in both women and men because it contains amino acids, fatty acids, flavonoids, saponins and sterols. Thus, this experiment investigated the effect of DPP supplementation on the oviposition rate, egg quality, ovarian follicle development, hematological variables and hormonal profile of laying hens. The findings indicated that the egg weight (EW), egg mass, albumen quality and laying rate of laying hens treated with DPP levels were significantly increased compared to those of the hens in the control group. The increased DPP levels significantly (*p* < 0.05) increased white blood cells (WBC), hemoglobin (Hb) and total antioxidant capacity (TAC), while H/L ratio significantly decreased. The increased DPP level significantly (*p* < 0.05) increased follicle-stimulating hormone (FSH) and luteinizing hormone (LH) concentration, ovary and oviduct weights compared to those of the control group.

**Abstract:**

This experiment studied the effect of DPP supplementation in the laying hens’ diet on the ovarian follicle development, hematological variables and hormonal profile of laying hens. Eighty-four, 78-week-old, Lohman LSL hybrids layers were equally divided into four groups (4 groups × 7 replicates × 3 hens); hens in the 1st group were fed a basal diet (control), while those in the 2nd, 3rd and 4th groups, were fed on the same diet and supplemented with 1.25, 2.5 and 5.0 g DPP/kg diet. The results showed that the egg weight, egg mass and laying rate of laying hens treated with DPP levels were significantly increased compared to those of the hens in the control group. Egg weight, egg surface area, albumen quality and percentage of the yolk in treated hens significantly increased compared with controls. The increased DPP levels in laying hens‘ diet significantly (*p* < 0.05) increased WBC, Hb and TAC, while heterophil/lymphocyte (H/L ratio) significantly decreased. Additionally, the concentrations of FSH and LH and the weights of ovary and oviduct in the treated hens significantly (*p* < 0.05) increased compared with the control. In conclusion, the DPP supplementation in the hen diet significantly improved egg production, EW, H/L ratio, ovarian follicles, FSH and LH hormones concentrations.

## 1. Introduction

In the European Union countries, using antibiotics as growth promoters in poultry farms, which were used as protection against the infectious diseases leading to increase economic losses in the poultry industry, has been banned [1]. Many investigators have tried to use some natural products for poultry nutrition in different farms to improve their production [2,3]. Date palm pollen (DPP—*Phoenix dactylifera* L.) is a natural product produced from palm trees grown in the different Arabian regions and collected by a human at the end of spring season, which amounted to approximately 1000 tons DPP every year [4]. Date palm pollen consists of 31.11% crude protein, 20.74% crude fat, 1.37% crude fiber, 13.41% carbohydrate, 28.80% moisture and 4.57% ash, as well as 57.9 mg essential oil/g total phenolic content [5,6].

Many investigators studied the physiological role of DPP against a variety of bacteria and viruses [7,8], anti-inflammatory and antiproliferative activities [9], antidiabetic effect [10] and antioxidants [11]. Studies also showed that the DPP contains estrogen, is and considered to have the ability to stimulate follicle-stimulating hormone (FSH) and luteinizing hormone to promote ovulation in the ovaries [12]. The findings of Shanoon et al. [13] and Mousa et al. [3], illustrated that using DPP has a significant positive effect on egg production, egg weight and egg mass, and they recommended using date palm pollen in a poultry diet to enhance the production performance.

In contrast, the findings of Ibrahim et al. [14] showed that the egg weight for quail hens treated with 5 g DPP/kg diet was not affected compared with untreated quails in a control group. Additionally, the findings of Mousa et al. [15] showed that the dietary supplementations of Fayoumi laying hens with DPP had significantly improved the nutrient digestibility and immune responses. The findings of Refaie et al. [16] showed that the hemoglobin, serum total protein, globulin and tissues total antioxidant capacity for Fayoumi chicks fed 0.1% for both DPP and DPP extract/kg diet were significantly increased, while total lipids and malondialdehyde in the tissues were significantly decreased as compared with the control group. Furthermore, Shihab [17] found that using dietary DPP at 2, 4 and 6 g/kg diet significantly improved total protein and globulin compared with a control group. Additionally, they noted that the white blood cells (WBC) as well as red blood cells (RBC) were not affected for broiler fed diet containing 2, 6 or 4 g DPP/kg diet on 21 or 34 days. Concerning the DPP supplementation on ovary and oviduct weights, the results of Erhaem [18] indicated that laying hens orally treated with 300 mg DPP/kg body weight significantly increased ovary and oviduct weight as well as luteinizing hormone (LH) and FSH concentration compared with a control group.

Generally, using DPP in medical science is pronouncedly increased, while the use of DPP in poultry production is limited. Therefore, this study aimed to study the effect of DPP supplementation on the laying hens’ diets on productive performance, egg production, hematological parameters, ovarian functions and hormonal profile.

## 2. Materials and Methods

The experiment was carried out at the experimental poultry farm of the Sohag University and approved by the Ethical Committee of Faculty of Agriculture, Sohag University, Sohag, Egypt (resolution no. 23/2019 from 7 August 2019).

### 2.1. Date Palm Pollen Collection

The DPP was collected during the flowering period at the end of March month before pollination from male palm trees growing in the Sohag governorate.

### 2.2. Date Palm Pollen Gross Chemical Composition

Crude protein, crude fiber, crude fat, ash and total sugars for DPP grains were determined according to A.O.A.C, [19]. The concentrations of calcium (Ca), potassium (K), iron (Fe) and magnesium (Mg) were determined by using Atomic Absorption Spectrometry (AAS) according to Beaty and Kerber [20].

### 2.3. Birds, Experimental Design and Diets

A total of 84 Lohman LSL hybrids layers, 78 weeks old (the late stage of the egg production period), were used in this experiment which lasted 6 weeks. All hens were equally assigned to four groups, with 7 replicates of 3 hens each (21 laying hens per each group). The laying hens were fed on a basal diet (mash form) to exceed requirements by the NRC, [21] (Table 1). In the 1st group, the hens were fed on a basal diet (control group), while those in the different treatment (2nd, 3rd and 4th groups) were supplemented with 1.25, 2.5 and 5.0 g DPP/kg diet, respectively. Fresh tap water was provided ad libitum during the experimental period. Each of the 3 birds were housed in a cage (42 width × 50 length × 40 height (cm)) and equipped with feeders and automatic nipples. The photoperiod during the experiment was set at 16L: 8D. The mean farm temperature was 24 ± 3 °C, and the relative humidity (RH) was 50–60%. The gross composition of DPP grains is presented in Table 2.

### 2.4. Production Parameters

The initial and final body weights were measured by using a digital balance at ±0.5 g at 78 and 84 weeks of age for each replicate within the treatment. The egg number and weight were recorded daily during the experimental period from 78 to 84 weeks of age. Egg weights were recorded daily, while egg mass (g/hen) was calculated by multiplying the laid eggs numbers and weight (g) for all replicates within each treatment. Feed consumption (g/hen/d) was recorded weekly for each replicate. Feed conversion ratio (FCR) was calculated as g feed/g egg.

### 2.5. Egg Quality Traits

At 82 weeks of age, external and internal egg quality traits were measured. One hundred and forty normal eggs were randomly collected from eggs laid in the last three days (4 groups × 7 replicates × 5 eggs) to determine the external and internal egg quality traits. Egg weight, shape index, egg surface area, shell weight and percentage were determined for egg quality measuring. Egg length (long axis) and egg width (short axis) were measured to determine shape index with the electronic caliper. Shape index (ShI %), Yolk index (YI %), Unit surface shell weight (USSW mg/cm^2^) and shell (%) were measured by the following equations according to Reddy et al. [22] and Anderson et al. [23]. ShI (%) = (Egg width/Egg height) ×100, YI (%) = (Yolk height/Yolk diameter) ×100, Egg surface area (cm^2^) = 3.9782 × egg weight^0.7056^, USSW (mg/cm²) = Egg weight (mg)/Egg surface area (cm^2^), Shell (%) = (Shell weight/Egg weight) ×100, Yolk diameter (mm) along the chalazae line was determined with the caliper. The albumen weight (g) was calculated from the difference between the entire egg weight and the yolk and eggshell weight. The eggshell was dried and weighed to the nearest 0.01 g after removal of the egg content.

### 2.6. Blood Constituents

At 82 weeks of age, 28 blood samples were collected from the brachial vein into heparinized tubes at 12 PM from 7 fasted birds per each group. All birds were fasted to 6 h before blood collection. Blood plasma was obtained by centrifugation for 20 min at 3000 rpm and stored at −20 °C until analysis and then allowed to thaw at room temperature (20 °C) before analysis. Total protein and albumin levels were measured by using commercial kits as per the manufacturer’s recommendations using a Biochemistry Auto Analyzer (Sinnowa D280, Nanjing, China).

The concentration of plasma MDA was determined by a commercial colorimetric kit (malondialdehyde; Biodiagnostic Co., Cairo, Egypt) according to Satoh [24], while total antioxidant capacity was measured using a colorimetric kit (Biodiagnostic Co., Cairo, Egypt). The total lipids contents were estimated as described by Bligh and Dyer [25].

### 2.7. Hematological Variables and Hormonal Profile

The counts of red blood cell (RBC) and white blood cells (WBC) were measured by using a hemocytometer, while the concentration of hemoglobin was measured by a hemoglobin meter. The heterophils, lymphocytes, eosinophils, basophils and monocytes were differentially counted as described by the method according to Gross and Siegel [26]. The H/L ratio was calculated by counting 100 cells and dividing heterophils numbers by lymphocyte numbers.

The plasma FSH and LH concentrations were measured using the commercially available ELISA kits, (CUSABIO, Wuhan, China), according to the manufacturer’s instructions for the respective kits.

### 2.8. Ovary, Oviduct Weights and Follicle Numbers

At the end of experiment, 28 laying hens were slaughtered (4 groups × 7 replicates × 1 hen/ replicate) to assess the productive and reproductive organs. Ovary and oviduct were removed and weighted separately to the nearest gram. The weights of ovary and oviduct were expressed as a live weight percentage. The diameter of the largest 1st, 2nd, 3rd, 4th and 5th follicle and Total Yolk Follicle (TYF) were measured by using a digital caliper within ±0.01 mm [27]; the different follicles were classified into 4 categories as follows: Small White Follicle = SWF (1–3 mm), Large White Follicle = LWF (3–5 mm), SYF = Small Yellow Follicle (SYF, 5–10 mm), LYF = Large Yellow Follicle (>10 mm), TYF = Total Follicles (1–10 mm). The follicle numbers in all groups at different categories were determined; and the largest follicles (F1–F5) were weighed by using a digital balance at ±0.01 g.

### 2.9. Statistical Analysis

The obtained data were subjected to a one-way analysis of variance with the treatment group effect using the general linear model procedure of SAS-6.03 [28] as follows: E_ij_ = μ + T_i_ + e_ij_, where: Y_ij_ = Observation measured, μ = Overall mean, T_i_ = Effect of date palm pollen treatment (1, 2, 3 and 4), E_ij_ = Random error component was normally distributed assumed. The significant differences between treatment means were determined by using Duncan [29]. The results were considered significantly different if *p* < 0.05 and tendencies were noted at *p*-values ≤ 0.10.

## 3. Results

### 3.1. Productive Performance

Findings concerning the effect of DPP supplementation on BW, egg weight, egg mass, egg production rate, as well as feed consumption and feed conversion ratio of laying hens supplemented with DPP at different levels, are presented in Table 3. The obtained findings showed that the averages of egg production rate of hens received DPP at different levels significantly (*p* < 0.05) increased relative to the control group. Additionally, DPP addition caused an increase in laying rate, and DPP supplementation had a significant effect on egg weight and egg mass. The obtained findings showed that egg mass pronounced by hens fed diets containing 1.25, 2.5 or 5.0 g DPP were significantly (*p* < 0.05) increased compared to those the control group.

In the present study, the final body weight and feed consumption of hens fed the diets contained DPP was not affected compared with control. The feed conversion ratio of hens in the treated group significantly (*p* < 0.05) improved as compared with the control group.

### 3.2. External and Internal Egg Quality Traits

Regarding egg quality traits (Table 4), the means of EW, egg surface area and USSW were significantly increased by DPP treatments compared with those in control group. Albumen height and percentage in treated hens were significantly (*p* < 0.01) increased, while yolk (%) was significantly (*p* < 0.05) decreased by DPP supplementation. Shape index, shell percentage and yolk index were not affected by DPP supplementation.

### 3.3. Blood Proteins, Malondialdehyde, Total Lipids and Total Antioxidant Capacity

Data presented in Table 5 revealed that the averages of plasma total protein, albumin, globulin, malondialdehyde and total lipids in the hens were not affected by DPP supplementation.

Referring to total antioxidant capacity (TAC), the obtained results indicated that the means of TAC in the hens treated with DPP were significantly (*p* < 0.01) increased, while the means of malondialdehyde and total lipids were not affected than those of the control.

### 3.4. Hematological Parameters, WBCs Differentia and Hormonal Profile

Results presented in Table 5 indicated that the RBC (×10^6^) in the treated hens was not as affected as that of the control one. On the other hand, the means of hemoglobin level and WBC count and lymphocyte in the treated hens significantly (*p* < 0.05) increased, while H/L ratio was significantly decreased compared to that of the hens in the control one.

From data illustrated in Table 5, it could be noticed that the DPP supplementation significantly (*p* < 0.05) increased FSH and LH concentrations in treated laying hens compared with the control group.

### 3.5. Liver, Spleen, Ovary, Oviduct and Ovarian Follicle Weights

The findings illustrated in Table 6 showed that supplementing LSL hens with DPP at different levels insignificantly increased the final body weight and liver percentage compared to that of the control group. On the other hand, spleen weight and percentage in treated hens significantly increased compared to that of the control.

With regard to ovary and oviduct weights, the obtained findings indicated that the supplementation of laying hens with DPP at different levels significantly increased the ovary and oviduct weights as compared with the control group.

From data presented in Table 7, the obtained data indicated that the numbers of SWF, LWF, SYF and TFN in the hens treated with DPP at different levels significantly increased compared to those of the control group, while the LYF numbers were not affected.

Referring to the largest yellow follicle weights (LYFW/g), the obtained data indicated that the first and fifth yellow follicle weights in the treated hens were significantly increased compared to the control, while the total LWF weight were significantly increased.

Regarding the largest yellow follicle diameters (LYFD/mm), the obtained data indicated that the second yellow follicle diameter (F_2_) in the treated hens was significantly increased, while the first (F_1_), third (F_3)_, fourth (F_4_) and fifth (F_5_) yellow follicles were not affected compared with the control.

## 4. Discussion

Briefly, the obtained results showed a significant improvement in egg production for treated hens; this improvement may be due to improved FSH and LH, which coincided with a high concentration of estradiol and estrogen hormone in DPP. This also may be due to the ability to increase oviduct and ovary growth as well as improve their functions [18]. The obtained findings showed that egg mass pronounced by hens fed diets containing 1.25, 2.5 or 5.0 g DPP were significantly (*p* < 0.05) increased compared with those in the control. This improvement in egg mass of the treated hens may be due to the improved egg number and feed conversion ratio compared with the control one. These results agreed with Mousa et al. [15], who noted that the averages of egg numbers and egg production rate in the Fayoumi laying hens treated with palm pollen were significantly increased compared to those of the control.

The results showed that the feed consumption of hens fed the diets containing DPP was not affected compared to that of the control. These results agreed with those of Refaie et al. [16], who found that feed consumption of Fayoumi chicks during a growing period was not affected by DPP supplementation. Similar results were also found by Batista et al. [30], who found that the feed consumption of broiler chicks treated with flavonoid was not as influenced as that of the control.

The significant improved feed conversion ratio of treated hens may be due to the development of the nutrient utilization throughout a beneficial microbial environment in the gut, which reflects high flavonoids content in the DPP. Additionally, this improvement may be due to increased egg mass in the treated hens compared with the control group. This finding is in harmony with those of Shanoon et al. [13], who concluded that the laying hens supplemented with DPP had a significantly improved feed conversion ratio compared to that of the control one.

The significant increase in the egg weight for treated hens may be attributed to the high estrogen hormone level, as the estrogen hormone promotes the oviduct growth and helps to form proteins for the oviduct and stimulates its formation. These findings agreed with those of Arhaem [31], who noted that the addition of DPP extract in the water recorded a significant difference in the egg production, egg weight and ovary tract weight compared to the control group. These results disagreed with those of Shanoon et al. [13], who found that egg quality characteristics were not affected (*p* < 0.05) except for shell weight and thickness.

Referring to plasma total protein, albumin, globulin, malondialdehyde and total lipids (Table 5), the insignificant improved total protein in treated hens may be attributed to the improvement in crude protein synthesis and digestion due to DPP supplementation. However, the increased globulin level in treated hens with increasing DPP levels may be due to the improved hen’s immunity, which reflects better liver efficacy in synthesizing enough globulins for immunologic action. These findings disagreed with those of Ibrahim et al. [14], who found that adding the mixture of 5 g Date palm pollen +0.5 g Panax Ginseng to the laying quail diets increased serum total protein significantly (*p* < 0.01) compared with the control.

Regarding the total antioxidant capacity (TAC), the significant increase in TAC in the treated hens may be attributed to high bioactive volatile unsaturated fatty acid contents as well as flavonoid compounds that play an important role as potent antioxidants, and have nutritional and physiological uses as dietary supplements to promote health. These results agreed with Refaie et al. [16], who found that the averages of total antioxidant capacity in hens treated with 1% DPP and DPP extract significantly (*p* < 0.05) increased by about 5.05% and 7.03%, respectively, compared with those in control group.

Table 5 showed a significant (*p* < 0.05) increase in hemoglobin level in the treated hens; this improvement may be due to the role of DPP in RBC membrane protection as well as increased iron level and its absorption from the digestive tract. Furthermore, the improved WBCs count, lymphocyte and H/L ratio in the treated hens supplemented with DPP may be due to minerals; antioxidant contents represented in the flavonoids and vitamins such as B1, B2 and B12, which consequently enhance the immune system in treated hens. Additionally, the increased lymphocytes percentage in the hens fed diets supplemented with DPP could be attributed to the improvement of their immunity system. These results agreed with Abuoghaba et al. [32], who found that hemoglobin level, lymphocytes, heterophils and H/L ratio in the chicks treated with bee pollen were significantly affected (*p* < 0.05), while monocytes and eosinophil were not affected.

From data illustrated in Table 5, the enhancement of serum FSH and LH concentrations in treated laying hens could be attributed to increased pollen content of gonadotropic and steroidal compounds [33], which play a role in the improvement of follicles development and ovulation. These results agreed with those of Hammed et al. [12], who found that FSH and LH concentrations of female rats treated with DPP extract grains increased significantly compared to those of the control. Similar results were also found by Akpan and Anietie [34], who stated that the FSH and LH concentrations were significantly (*p* < 0.05) improved in rats treated with herbal extraction that has the same active ingredients of DPP, such as Tetracarpidium conophorum, which contains proteins, carbohydrates, tannins, oils, vitamins and minerals. LH plays critical roles in follicular development and consequently ovulation; in addition to, it can promote granulosa cells to secrete progesterone prior to follicle ovulation [34].

The significant increase in spleen weight and percentage in treated hens may be due to the increased spleen activity and efficiency to improve the production of white blood cells compared to the hens in the control group. These results agreed with the findings of Nady et al. [35], who reported that the spleen weight of mice orally treated for 14 days simultaneously with 1 mg DPP/kg body weight significantly increased as compared to the control group.

With regard to ovary and oviduct weights, the significant increase in the ovary and oviduct weights for the hens treated with DPP may be due to increased FSH and LH secretions, which in turn encourage the development and numbers of the follicles compared to the control group. Furthermore, the increased ovary and oviduct weight in treated hens may be attributed to the increased estrogen in DPP, which consequently increased the growth and development of the oviduct and the integration of functions and also increased their cell numbers.

Likewise, the increased oviduct weight of hens treated with DPP may be due to high estrogen concentration, which promotes the oviduct growth and helps to make the oviduct proteins. These findings agreed with Erhaem [18], who noted that the ovary and oviduct weights for laying hens orally treated daily with 200 and 300 mg DPP/kg BW significantly increased compared to that of the control group. Additionally, this improvement in the relative weight of the oviduct in the treated hens may be due to many important vitamins and nutrients in DPP such as vitamins E, A and B [5]. Similarly, the findings of Dan Shao et al. [36] found that the supplementation of laying hens with daidzein (DA) as a natural product extracted from soy plants improved luteinizing hormone (LH) levels and small yellow follicle (SYF) numbers.

The significant increase in numbers of SWF, LWF, SYF and TFN for the hens treated with DPP at different levels could reflect the superiority of date palm pollen treatments in the number and diameter of the ovarian follicles; this reflects that the pollen contains mainly phenolic, flavonoids and carotenoids as well as estrogen, which has a role in increasing the activity of FSH and LH, leading to an increase in the number and maturity of the follicles [37]. These findings agreed with those of Al-Salhie et al. [2], who found that the number and diameter of the primary ovarian follicle for Japanese quail females treated with 1000, 750, 500 and 250 mg DPP at 60 days increased significantly compared to that of the control group, since the highest number (18) was recorded in hens treated with 1000 mg DPP, while the lowest one (8) was recorded in the control. Furthermore, the findings of Dan Shao et al. [36] showed a significant (*p* < 0.05) increase observed in small yellow follicle numbers in the hens treated with daidzein. However, the numbers of pre-ovulatory follicle, atresia follicle and big white follicle were not affected relative to the control group.

Referring to the largest yellow follicle weights (LYFW/g), the significant increase in the first and fifth yellow follicle weight in the treated hens agreed with those of Ebeid et al. [27], who noted that the LYF numbers and total weights were not influenced by fish oil treatment, while the largest follicle (F1) weight was significantly increased in the hens treated with 2.5% fish oil compared to in other treatments. Additionally, they added that the SYF number was significantly (*p* < 0.05) affected by fish oil treatment, while the LWF numbers were not changed by fish oil supplementation.

Regarding the largest yellow follicle diameters (LYFD/mm), the significant increase in the second yellow follicle diameters in the treated hens agreed with those of Oguike et al. [38], who found a slower follicular maturation rate in the ovaries of aged hens than those of the young ones.

## 5. Conclusions

In summary, the obtained data showed that dietary DPP supplementation in laying hen diets significantly improved laying reproductive performance. The LSL hens treated with DPP had significantly improved laying rate, egg mass and H/L ratio as well as follicle numbers and FSH and LH concentrations. The DPP supplemented diet significantly improved ovary and oviduct weight as well as total antioxidant capacity in laying hens. Using DPP in a 2.5 g/kg diet for laying hens is more economic, particularly for small farmers and producers. The current findings provide a beneficial reference for DPP applications as a natural product to improve the laying rate of laying hens.

## Figures and Tables

**Table 1 animals-11-00069-t001:** Composition and calculated analysis of layer diet during experimental period (as-fed basis).

Ingredients (%)	%	Calculated Nutrients	%
Soybean meal (44%)	24	Crude protein	16.50
Yellow corn	61.57	Metabolizable energy (kcal/kg)	2700
Wheat brain	6.7	Crude fiber	3.5
Corn gluten 60%	4.5	Crude fat	3.0
NaCl	0.37	Calcium	3.4
Limestone	1.16	Available phosphorus	0.40
Di-Calcium Phosphate	1.39	Lysine	0.70
Vitamins and Minerals premix *	0.30	Methionine	0.34
DL-Methionine	0.01	Met. + Cyct	0.62
Total	100	Sodium	0.16

* Each 1 kg of vitamins and minerals premix included: vitamin A 10000 I.U; vitamin E 15 mg; vitamin D3 2000 I.U.; vitamin B1 1 mg; vitamin K3 1 µg; vitamin B12 10 µg; vit. B2 5 mg; vit B6 1.5 mg; pantothenic acid 10 mg; niacin 30 mg; biotin 50 mg; folic acid 1 mg; choline 300 mg; copper 4 mg; iodine 0.3 mg; zinc 50 mg; iron 30 mg; manganese 60 mg; selenium 0.1 mg; CaCo3 up to 1 kg, cobalt 0.1 mg and carrier.

**Table 2 animals-11-00069-t002:** Gross chemical composition of date palm pollen grains.

Contents	g/100 g	Minerals	mg/100 g	%
Crude protein	36	Ca	530	0.0053
Energy (kcal)	315	K	760	0.0076
Crude fiber	8.8	Mg	310	0.0031
Crude fat	11.8	Fe	225	0.00225
Ash	9.26	Zn	125	0.00125
Carbohydrate	17.1	Mn	310	0.0031

**Table 3 animals-11-00069-t003:** Effect of date palm pollen (DPP) supplementation on body weight, egg mass, egg production rate and feed utilization of laying hens.

Parameters	DPP (g/kg Diet)				SEM	*p*-Value	Significance
	Control	1.250	2.50	5.0			
IBW (g)	1623.80	1628.57	1623.81	1628.57	36.77	0.9994	NS
FBW (g)	1420.71	1453.33	1458.81	1461.90	38.12	0.7297	NS
**Egg Weight (g)**							
78–80 weeks	68.57	69.98	71.01	71.43	1.17	0.3388	NS
80–82 weeks	68.95 ^b^	69.82 ^a,^^b^	71.64 ^a^	71.45 ^a^	0.77	0.0590	*
82–84 weeks	69.58 ^a^	70.21 ^a^	71.34 ^a^	71.70 ^a^	0.86	0.2935	*
78–84 weeks	69.03 ^b^	70.00 ^a,b^	71.33 ^a^	71.53 ^a^	0.66	0.0432	*
**Egg Mass (g)**							
78–80 weeks	812.14 ^b^	981.10 ^a,b^	1048.86 ^a^	1029.28 ^a^	61.41	0.0469	*
80–82 weeks	857.00 ^b^	1027.50 ^a^	1072.75 ^a^	1061.19 ^a^	52.05	0.0253	*
82–84 weeks	912.86 ^b^	1053.60 ^a,b^	1112.04 ^a^	1083.99 ^a,b^	57.84	0.0989	*
78–84 weeks	860.67 ^b^	1020.66 ^a^	1077.88 ^a^	1058.16 ^a^	40.27	0.0033	*
**Egg Production (%)**							
78–80 weeks	56.46 ^b^	66.67 ^a,b^	70.75 ^a^	68.71 ^a,b^	4.31	0.0118	*
80–82 weeks	59.18 ^b^	70.07 ^a^	71.43 ^a^	70.75 ^a^	3.52	0.0670	*
82–84 weeks	62.59 ^b^	71.43 ^a^	74.15 ^a^	72.11 ^a^	3.95	0.1972	*
78–84 weeks	59.41 ^b^	69.39 ^a^	72.11 ^a^	70.52 ^a^	2.62	0.0092	*
**Feed Consumption (g)**							
78–80 weeks	1571.90	1584.19	1569.69	1586.18	32.53	0.9771	NS
80–82 weeks	1647.92	1673.70	1656.31	1679.80	26.12	0.8088	NS
82–84 weeks	1660.06	1701.93	1705.98	1711.09	32.83	0.7618	NS
78–84 weeks	1628.63	1653.27	1643.99	1659.02	23.66	0.8149	NS
**Feed Conversion Ratio (g feed/g egg)**							
78–80 weeks	1.94 ^a^	1.63 ^b^	1.59 ^b^	1.57 ^b^	0.10	0.0567	*
80–82 weeks	1.95 ^a^	1.66 ^b^	1.58 ^b^	1.60 ^b^	0.10	0.0461	*
82–84 weeks	1.94 ^a^	1.68 ^b^	1.62 ^b^	1.60 ^b^	0.07	0.0060	*
78–84 weeks	1.90 ^a^	1.63 ^b^	1.56 ^b^	1.57 ^b^	0.07	0.0043	*

^a,b^ Means with at least one common superscript in a row do not differ significantly (*p* > 0.05). IBW (g) = initial body weight, FBW (g) = final body weight. * Significant difference at 0.05; NS = non significance at 0.05.

**Table 4 animals-11-00069-t004:** Effect of DPP supplementation on egg quality traits of laying hens.

Parameters	DPP (g/kg Diet)				SEM	*p*-Values	Significance
	Control	1.25	2.5	5.0			
**External Egg Quality**							
Egg weight (g)	68.30 ^b^	69.73 ^a,b^	71.57 ^a^	71.46 ^a^	0.75	0.0058	**
Shape index (%)	73.54	73.64	72.92	72.17	0.80	0.0346	NS
Surface area (cm^3^)	78.32 ^b^	79.47 ^a,b^	80.94 ^a^	80.88 ^a^	0.60	0.0058	**
USSW (mg/cm^2^)	87.13 ^b^	87.66 ^a,b^	88.33 ^a^	88.32 ^a^	0.28	0.0058	**
Shell (%)	11.95	11.67	11.35	11.40	0.27	0.3844	NS
**Internal Egg Quality**							
Albumen height (mm)	6.84 ^b^	7.18 ^a,b^	7.48 ^a^	7.17 ^a,b^	0.17	0.0546	*
Albumen (%)	55.22 ^b^	56.71 ^a,b^	57.77 ^a^	57.59 ^a^	0.63	0.0197	*
Yolk-index (%)	30.23	31.05	31.61	31.78	1.02	0.7064	NS
Yolk (%)	32.83 ^a^	31.61 ^a,b^	30.87 ^b^	31.01 ^b^	0.54	0.0414	*

^a,b^ Means with at least one common superscript in a row do not differ significantly (*p* > 0.05). * Significant difference at 0.05; ** Significant difference at 0.01; NS = non significance at 0.05.

**Table 5 animals-11-00069-t005:** Effect of DPP supplementation on the blood parameters, hematological estimates and hormonal profile of laying hens.

Parameters	DPP (g/kg Diet)				SEM	*p*-Values	Significance
	Control	1.25	2.50	5.0			
**Blood Parameters**							
Total protein (g/dL)	3.58	4.24	4.38	4.42	0.40	0.3289	NS
Albumin (g/dL)	1.39	1.33	1.63	1.60	0.16	0.5013	NS
Globulin (g/dL)	2.20	2.82	2.96	2.82	0.42	0.5895	NS
MDA (mmol/100 g)	1.77	1.45	1.45	1.44	0.19	0.5634	NS
Total lipids (mg/100 g)	530.29	460.29	461.86	455.14	26.72	0.1767	NS
TAC (mmol/100 g)	31.79 ^b^	38.71 ^a^	38.43 ^a^	38.57 ^a^	1.68	0.0181	*
**Hematological Parameters**							
RBC (×10^6^)	2.72	3.32	3.22	3.35	0.36	0.5759	NS
WBC (×10^3^)	14.79 ^b^	18.29 ^a,b^	17.71^ab^	18.57 ^a^	1.17	0.0153	*
HB (g/dL)	9.63 ^b^	12.11^a^	12.43 ^a^	12.57 ^a^	0.67	0.0155	*
**WBCs Differentia**							
Heterophils (%)	58.29	55.71	55.00	55.86	1.29	0.3175	NS
Lymphocyte (%)	33.71 ^b^	38.29 ^a,b^	39.43 ^a^	39.14 ^a^	1.58	0.0591	*
Basophil (%)	1.29	0.86	1.29	1.14	0.49	0.9162	NS
Eosinophil (%)	4.57	3.29	2.86	2.71	0.78	0.3454	NS
Monocyte (%)	2.14	1.86	1.43	1.14	0.50	0.5147	NS
H/L ratio	1.78 ^a^	1.48 ^a,b^	1.40 ^b^	1.44 ^b^	0.10	0.0509	*
**Hormonal Profile**							
FSH (ng/mL)	13.87 ^b^	15.89 ^a,b^	16.97 ^a^	16.87 ^a,b^	0.80	0.0417	*
LH (ng/mL)	5.53 ^b^	8.01 ^a,b^	8.34 ^a^	8.41 ^a^	0.89	0.0543	*

^a,b^ Means with at least one common superscript in a row do not differ significantly (*p* > 0.05). * Significant difference at 0.05; NS = non significance at 0.05. MDA = malondialdehyde, TAC = total antioxidant capacity, RBC = Red blood cells (×10^6^), WBC = White blood cells (×103), HB (g/dL) = hemoglobin, FSH= follicle-stimulating hormone, LH = luteinizing hormone.

**Table 6 animals-11-00069-t006:** Effect of DPP supplementation on body weight, liver, spleen and some reproductive parameters of laying hens.

Variables	DPP (g/kg Diet)				SEM	*p*-Values	Significance
	Control	1.25	2.50	5.0			
Body weight (g)	1419.29	1451.71	1456.86	1459.14	47.60	0.9273	NS
Liver weight (g)	32.59	35.04	36.03	34.00	2.53	0.7972	NS
Liver (%)	2.309	2.407	2.485	2.371	0.17	0.8965	NS
Spleen weight (g)	1.086 ^b^	1.271 ^a,b^	1.343 ^a,b^	1.400 ^a^	0.09	0.0120	*
Spleen (%)	0.076 ^b^	0.088 ^a,b^	0.092 ^a,b^	0.096 ^a^	0.01	0.0089	*
Ovary (g)	29.943 ^b^	43.11 ^a^	43.53 ^a^	44.31 ^a^	3.76	0.0363	*
Ovary (%)	2.1261 ^b^	2.985 ^a^	2.996 ^a^	3.053 ^a^	0.27	0.0604	*
Oviduct (g)	41.54 ^b^	49.58 ^a,b^	50.30 ^a^	50.80 ^a^	2.86	0.0087	*
Oviduct (%)	2.97	3.43	3.46	3.55	0.25	0.3807	NS

^a,b^ Means with at least one common superscript in a row do not differ significantly (*p* > 0.05). * Significant difference at 0.05; NS = non significance at 0.05.

**Table 7 animals-11-00069-t007:** Effect of DPP supplementation on follicular numbers, diameters and weights of laying hens.

Parameters	DPP (g/kg Diet)				SEM	*p*-Values	Significance
	Control	1.25	2.50	5.0			
**Follicle Numbers**							
SWF (1–3 mm)	21.71 ^b^	26.29 ^a^	26.00 ^a,b^	27.14 ^a^	1.48	0.0684	*
LWF (3–5 mm)	10.86 ^b^	14.86 ^a^	14.57 ^a^	15.14 ^a^	0.93	0.0106	*
SYF (5–10 mm)	5.43 ^b^	7.71 ^a^	7.29 ^a,b^	8.14 ^a^	0.65	0.0354	*
LYF (>10 mm)	2.71	3.43	3.71	3.86	0.48	0.3678	NS
TFN (no.) 1–10 mm	40.86 ^b^	52.29 ^a^	51.57 ^a^	54.29 ^a^	2.05	0.0005	***
TFW (1–10 g)	20.11	27.13	26.71	29.77	3.27	0.2215	NS
**Largest Yellow Follicle Weights (LYFW/g)**							
F_1_ (g)	11.029 ^b^	13.071 ^a,b^	13.443 ^a,b^	14.286 ^a^	1.02	0.0170	*
F_2_ (g)	6.143	7.500	8.486	9.100	1.17	0.3272	NS
F_3_ (g)	2.114	3.714	3.029	4.157	0.97	0.4833	NS
F_4_ (g)	0.643	1.771	1.071	1.636	0.53	0.4254	NS
F_5_ (g)	0.186 ^c^	1.07 ^a^	0.686 ^a,b^	0.596 ^b,c^	0.15	0.0036	**
**Largest Yellow Follicle Diameters (mm)**							
F_1_ (mm)	20.64	21.86	24.43	25.43	1.60	0.1547	NS
F_2_ (mm)	15.57 ^b^	17.57 ^a,b^	20.57 ^a^	21.86 ^a^	1.48	0.0157	*
F_3_ (mm)	10.86	13.43	14.14	15.71	1.82	0.3196	NS
F_4_ (mm)	7.57	9.71	9.28	9.86	1.51	0.6975	NS
F_5_ (mm)	4.86	6.71	5.43	5.71	0.65	0.2606	NS

^a,b,c^ Means with at least one common superscript in a row do not differ significantly (*p* > 0.05). SMF = Small White Follicle (1–3 mm), LWF = Large White Follicle (3–5 mm), SYF = Small Yellow Follicle (SYF, 5–10 mm), LYF = Large Yellow Follicle (>10 mm), TYF = Total Follicles (1–10 mm). F_1_ = The first yellow follicle; F_2_ = The second yellow follicle; F_3_ = The third yellow follicle; F_4_ = The fourth yellow follicle; F_5_ = The fifth yellow follicle; * Significant difference at 0.05; ** Significant difference at 0.01; *** Significant difference at 0.001; NS = non significance at 0.05.

## Data Availability

All data sets obtained and analyzed during the experiment are available on fair request from the respective author.

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
