# Peer review of "Effect of Date Palm Pollen Supplementation on the Egg Production, Ovarian Follicles Development, Hematological Variables and Hormonal Profile of Laying Hens"

_animals, 2021, doi:10.3390/ani11010069_

Round 1

Reviewer 1 Report

The paper studied the effect of increasing levels of pollen supplementation on the reproductive performance and on some physiological traits of laying hens, from 78 until 84 weeks of age.

The paper shows many unclear descriptions and/or few details, and many mistakes (English language, typing..?) in the text, thus it does not allow an exhaustive and  final evaluation.

Some words concerning the studied traits should be changed (impact instead of effect, hormonal estimates instead of hormonal profile, ..).

I am indicating only some points of the paper, but it needs a deep total revision.

The abstract is unclear in many points (row 24, 25,33,..); how many weeks of treatment?

Introduction:

rows 45-47, 57-58, 66-67, unclear.

Materials and methods:

Details and description of the analyses have to be given in the text and not only references.

Why a study on the effect of pollen from 78 to 84 weeks? More explanations and discussion.

Which was the physical form of the pollen added to the diets? How it was added?

Table 1: ingredients: only one %; diet composition: as fed basis or d.m.?

Table 2: Gross composition??

Row 118:equal

Row 121- 128: more details!

Row 133: fasted? Details!

Row 141: details!

Row 146, 151: details!

Row 163: ??

Row 177, 180: egg number?

Table 3: and the body weight? In the title.

Why a final body weight lower than the initial body weight? In the text no comment regarding body weight data is shown.

In general, you should indicate the number of observations.

Row 197: USSA??

Table 4: you should change the row DDP…. Control.., and eliminate g in the second row

Surface area: g as unit?

In some tables, I do not understand some values or units of some traits!

Table 6: I suggest to change the title of this table , as the BW and liver.. are considered as reproductive parameters or involved with reproduction?

Row 246: not true for all the levels.

Row 258:??

Row 267: feed utilization : what do you mean?

Given that you tested increasing levels of DPP, why did you not test the linear or other components?

As written above, in the discussion you should give explanations of the importance of testing DPP in the diet at this laying phase, considering the physiological status of the hens in this period of their egg production activity.

Row 331-336: more comment regarding the body weight of the hens.

Row 345: why?

Row 348: cholesterol??

A discussion on the best DPP level should be given.

Author Response

Dear editor

Greetings, appreciation and thanks to the reviewers

First, I would like to refer to the amendment in the order of the names of authors in the manuscript to become 

  1. Saleh1, D. Kokoszyński2*, M. A. Mousa3 and A. A. Abuoghaba1

Second, Answers to the comments and suggestions of reviewers

REVIEW RAPORT 1

Comments and Suggestions for Authors

The paper studied the effect of increasing levels of pollen supplementation on the reproductive performance and on some physiological traits of laying hens, from 78 until 84 weeks of age.

Comment 1:

The paper shows many unclear descriptions and/or few details, and many mistakes (English language, typing..?) in the text, thus it does not allow an exhaustive and  final evaluation.

Answer:

All mistakes were corrected according the reviwers comments.

Comment 2:

Some words concerning the studied traits should be changed (impact instead of effect, hormonal estimates instead of hormonal profile).

Answer:

Impact instead of “effect” in line 73,

"Hormonal profile" was not found in the text 

Comment 3:

The abstract is unclear in many points (row 24, 25,33,..); how many weeks of treatment?

Answer:

Line 24, 25:

The sentence changed to be " This experiment studied the impact of DPP supplementation in the laying hens diet on ovarian follicle development, hematological variables, and hormonal estimates of laying hens "

Line 33:

The increased DPP levels significantly (P<0.05) increased WBS, Hb, and TAC, while H/L ratio significantly decreased

Answer:

L32-34

The sentence changed to be ‘The increased DPP levels in laying hens diet significantly                 (p < 0.05) increased WBS, Hb, and TAC, while H/L ratio significantly decreased’

Comment 4 "Introduction"

Lines 45-47:

The sentence " Date palm pollens (DPP-Phoenix dactylifera L.) is a natural product collected by a human from Date Palm males during the pollination period in palm trees at the end of spring season, which produced every year approximately, 1000 tons DPP are produced by millions of palm trees grown in the different Arabian region [4]"

Answer:

L44-47

The sentence changed to be "Date palm pollens (DPP-Phoenix dactylifera L.) is a natural product produced from palm trees grown in the different Arabian regions and collected by a human at the end of spring season, which amounted approximately 1000 tons DPP every year [4]."

Comment 5: (Line 57-58):

The sentence " In contrast, Ibrahim et al. [14] found that the average of egg weight for quail hens treated with 5g DPP/kg diet insignificantly affected compared to the control group."

Answer:

Answer:

L57-58

The sentence achanged to be " In contrast, the findings of Ibrahim et al. [14] showed that the egg weight for quail hens treated with 5 g DPP/kg diet was not affected compared with untreated quails in control group".

Comment 6: (Line 66):

The sentence", while albumin has not changed on day 21 between experimental groups"

Answer:

Delete the sentence", while albumin has not changed on day 21 between experimental groups" was

Comment 7: Line 67:

Answer:

Also, the who added that the WBC" delete "the" Also, who added that the WBC on day 21 for 2 and 6g/kg diet DPP as well as RBC and WBC for 4g/kg diet on day 34 were not influnced.” and rewrite sentence to be "Also, who added that the WBC for broiler fed diet containing 2 and 6g DPP /kg diet on day 21 as well as RBC and WBC for broiler fed diet 4g/kg diet on day 34 were not affected"

Comment 8: (Line 73):

Impact instead of “Effect”

Comment 9: (Materials and methods):

Why a study on the effect of pollen from 78 to 84 weeks? More explanations and discussion.

Answer:

This experiment was carried out at the end of egg production period, it is known that egg production chickens suffer from low egg production, especially after 70 weeks of age, so the experiment aimed to study the possibility of adding DPP to laying hens diets of to improve egg production and the related physiological characteristics

Comment 10:

Which was the physical form of the pollen added to the diets? How it was added?

Answer:

Date palm pollen was added in the powder form, and the suspension was mixed homogeneously to reach the appropriate shape for the fluff at different levels.

Comment 11:

Table 1: ingredients: only one %; diet composition: as fed basis or d.m.?

Answer:

As-fed basis

Comment 12:

Table 2: Gross composition??

Answer:

Gross chemical composition of Egyptian DPP grains.

Comment 13: (Line 118): Equal

Answer

"Equal" deleted from sentence

Comment 14: (Line 121- 128): more details!

Answer:

All parameters were measured according different methods as described in the text.

Comment 15: (Line 133): fasted? Details!

Answer:

L134

Changed to be "from 7 fasted birds per each group."

"Fasted" word not found in line 133

Comment 16: (Line 141): details!

Answer:

Comment 17: (Line 146, 151): details!

Answer:

Line 146: This method requires breaking down blood cells by treated with chemicals that rupture the red blood cell wall and combine with the hemoglobin to form a compound that can be measured photometrically. The result is displayed in digital form on the face of the instrument.

Line 151: FSH and LH concentrations were measured using the commercially available ELISA kits, CSB-E06867h for FSH, and CSB-E12690h for LH (CUSABIO, Wuhan, China), according to manufacturer’s instructions for respective kits. Standards and samples were assayed in a volume of 50 μl in duplicate. The minimum detectable concentration of FSH and LH using these kits is typically less than 1 mIU/ml and 0.5 mIU/ml, respectively.

Comment 18: (Line 163): ?? the largest follicles (F1-F5) were weighed.

Answer:

Rewrite to be (the largest follicles (F1-F5) were weighed by using a digital balance at ± 0.01 g.)

Comment 19: (Line 177, 180): egg number?

Answer

Egg number was removed in lines 177 and 180.

Comment 20: (Table 3: and the body weight? In the title).

Answer:

L185-186

Table 3. Impact of DPP supplementation on body weight, egg mass, laying rate and feed utilization of laying hens.

Comment 21:

Why a final body weight lower than the initial body weight? In the text no comment regarding body weight data is shown.

Answer:

The decreased final body weight due to the aging of the hens, and it has begun to enter the molting stage. No comment regarding body weight data is shown due to insignificant effect in the body weight and inserted in line 191 to be” the final body weight and feed consumption of hens”

Comment 22: (Line 197): USSA??

Answer:

USSA = unit shell surface area, this scale is used as an indicator of egg size and weight.

Comment 23:

DPP supplementation (g/kg diet) to be DPP (g/kg diet) and delete ‘g” in the second row in all tables.

Comment 24:

Surface area: g as unit?

Answer:

 cm3 as unit

Comment 25:

Table 6: I suggest to change the title of this table , as the BW and liver.. are considered as reproductive parameters or involved with reproduction?

Answer:

L285-286

Table 6. Impact of DPP supplementation on BW, liver, spleen, liver and some reproductive parameters of laying hens.

Comment 26: (Line 246): not true for all the levels "while the total LWF weight were insignificantly increased"

Answer:

while the total LWF weight were significantly increased.

Comment 27: (Line 258):??

Answer:

L257-259

The sentence rewrite to be “Regarding largest yellow follicles diameters (LYFD/mm), the obtained data indicated that the second yellow follicle diameter (F2) in the treated hens were significantly increased, while the first (F1, third (F3), fourth (F4) and  fifth (F5) yellow follicle were not affected compared with control”.

Comment 28: (Line 267): feed utilization: what do you mean?

Asswer:

Feed utilization changed to be (feed conversion ratio)

Comment 29:

Given that you tested increasing levels of DPP, why did you not test the linear or other components?

Answer:

The high DPP levels have been tested to study its effect on reproductive and reproductive performance. Other ingredients have not been tested because we only have palm pollen, and other ingredients will be studied in the future.

Comment 30:

As written above, in the discussion you should give explanations of the importance of testing DPP in the diet at this laying phase, considering the physiological status of the hens in this period of their egg production activity.

Answer:

In the discussion, we given explanations of the importance of testing DPP in the diet at this laying phase in relation to physiological status of the hens in this period of their egg production activity.

Comment 31: (Line 331-336):

more comment regarding the body weight of the hens.

Answer:

The final body weight was insignificantly influnced by DPP supplementation, thus the authors not discus this parameter, we insert body weight in line 191.

Comment 32: (Iine 345): why?

Answer:

This results were found by Dan Shao et al. [36], who noted that the supplementation of laying hens with daidzein (DA) as natural product extracted from soy plants improved luteinizing hormone (LH) levels, and small yellow follicle (SYF) numbers.

Comment 33: Line 348: cholesterol??

Answer:

Phenolics and flavonoids insted of cholesterol.

Comment 34:

A discussion on the best DPP level should be given.

Answer:

In conclusion, the authors concluded that Using DPP at 2.5 g/kg diet for laying hens is more economic particularly for small farmers and producers.

After reading the text, changes in accordance with the instructions for authors are marked in blue.

Reviewer 2 Report

The manuscript of Saleh et al. increases the knowledge on the DPP supplementation on the avian reproductive system, hematological variables, and hormonal estimates. In general manuscript is well written. The methods are correct and well described. The results of this study were described in details. Within the discussion there is a lack of reference 33. The reference style should be unified.  

Author Response

REVIEW RAPORT 2

Comment 1:

Within the discussion there is a lack of reference 33. The reference style should be unified.  

Answer:

Reference [33] is included in the discussion

Reviewer 3 Report

I think this paper is worthy of publication with appropriate English language corrections. It is a good study that provides confirmatory information that should improve egg production practices. There are many corrections needed- inappropriate capitalizations, run on sentences, comma splices, etc that must be corrected by someone expert in English. Also, I would write out the abbreviations in the abstract for clarity. Reference 4 has 2 authors with the same name. Is this correct?

Author Response

REVIEW RAPORT 3

Comment 1:

There are many corrections needed- inappropriate capitalizations, run on sentences, comma splices, etc that must be corrected by someone expert in English. Also, I would write out the abbreviations in the abstract for clarity. Reference 4 has 2 authors with the same name. Is this correct?

Answer:

Many required corrections were made, taking into account inappropriate capitalization, running sentences, and comma breaks.

Reference 4 has only one author with the same name.

  1. Mehraban, F.; Jafari, M.; Toori, M.A.; Sadeghi, H.; Joodi, B., Mostafazade, M.;   Sadeghi, H. Effects of date palm pollen (Phoenix dactylifera L.) and Astragalus ovinus on sperm parameters and sex hormones in adult male rats. Iran. J. Reprod. Med. 2014, 12, 705-712. PMID: 25469129; PMCID: PMC4248157).

Round 2

Reviewer 1 Report

I appreciate the work of the authors for improving the paper following suggestions, but probably many suggestions were not completely understood and thus many parts of the paper needs a further revision.

Some sentences are not clear, may be also  for typing mistakes.

In the text many inadequate expressions such as "egg mass was heavier" or "was insignficantly affected" and Others have to be corrected.

I suggested to avoid the words " impact" and "hormonal estimates", but they were not changed in "effect" and "hormonal profile".

To the title of the paper should be added oviposition rate , egg quality, or egg production (check the number of words)

 row 15: an initial sentence for justifying the use of DPP

row 16: oviposition rate and egg quality

row 18: and albumen quality

same addition to the abstract

row 37 improved egg production, EW..

row 62-63: not clear

row 66: not clear

row 69: not clear

row 85: why gross chemical composition?

I suggest for DPP composition (table 2) centesimale and mineral composition (d.m. basis?)

How did you measured the energy content?

for the table 1: how did you calculated  nutrients? you should indicate reference.

why minerals with words and symbols? in the text ok, but not in tables.

row 106: not clear

row 122-129: check all very well,many mistakes.

row 134: how many hours of fasting?

row 155: check it

row 158-159:not clear

row 168: not clear

row 176: ..supplementationon egg weight, ...and the other traits?

table 4: surface area: how it was measured?

USSA or USSW?

yolk index: units?

All the variables needs a comment  on the effect.

row 228: other variables in the tables.

These are the main suggestions, but all the paper, each part needs revision.

Author Response

Comments and Suggestions for reviewers

Comment 1:

In the text many inadequate expressions such as "egg mass was heavier" or "was insignificantly affected" and Others have to be corrected.

Answer:

Significantly heavier corrected to be (significantly increased) in lines 183, 266, 361

Insignificantly affected changed to be (not affected) in lines 191, 216, 220, 245, 272, 273, 275, 360, 363,

Comment 2:

I suggested to avoid the words “impact" and "hormonal estimates", but they were not changed in "effect" and "hormonal profile".

Answer:

Impact changed to be “Effect” in lines 2, 15, 24, 73, 176, 185, 200, 208, 235, 250

Hormonal estimates changed to be “hormonal profile” in lines 4, 17, 25, 38, 75, 144, 209, 218,  

Comment 3:

To the title of the paper should be added oviposition rate, egg quality, or egg production (check the number of words)

Answer:

Oviposition changed to be egg production in lines 3, 16

Comment 4:

 Row 15: an initial sentence for justifying the use of DPP

Answer:

Date palm pollen (DPP) is a natural product produced from male palm flowers and using to improve ovulation and fertility in both women and men because it contains amino acids, fatty acids, flavonoids, saponins and sterols.

Comment 5:

Row 16: oviposition rate and egg quality

Answer:

oviposition rate, egg quality were added in line 16

Comment 6:

Row 18: and albumen quality

same addition to the abstract

Answer:

Albumen quality was added in lines (20, 33) in simple summary and abstract.  

Comment 7:

Row 37 improved egg production, EW..

Answer:

Egg production was added in line 38.

Comment 7:

Row 62-63: not clear

Answer:

The authors compared between DPP and DPP extract supplementation on studied traits.

The sentence corrected to be “The findings of Refaie et al. [16], showed that the hemoglobin, serum total protein, globulin and tissues total antioxidant capacity for Fayoumi chicks fed 0.1% for both DPP and DPP extract/ kg diet were significantly increased, while total lipids and malondialdehyde in the tissues were significantly decreased as compared with the control group.”

Comment 8:

Row 66: not clear

Answer:

The sentence rewrite to be” Also, who noted the WBC was well as RBC and WBC were not affected for broiler fed diet containing 2, 6 or 4g DPP/kg diet on 21 or 34 days).

Comment 9:

Row 69: not clear

Answer:

The sentence rewrite “Indicated that laying hens orally treated with 300 mg DPP/kg BW significantly increased ovary and oviduct weight as well as LH and FSH concentration compared with control group”.    

Comment 10:

Row 85: why gross chemical composition?

Answer:

This method is only available in my lab.

Comment 11:

I suggest for DPP composition (table 2) centesimale and mineral composition (d.m. basis?)

Answer 11:

All ingredients were analyzed with 100g of palm pollen

The minerals percentage was calculated for 100g of DPP

Comment 12:

How did you measure the energy content?

Answer:

Energy was estimated according to NRC equation on energy NRC [21] 

Comment 13:

For the table 1: how did you calculated nutrients? You should indicate reference.

Answer:

The requirement of this breed is designed according to the breed catalog https://www.ltz.de/de-wAssets/docs/management-guides/en/Cage/White/LTZ-Management-Guide-LSL-Lite.pdf.

There are many plates containing calcium carbonate (limestone) as free choice to birds.

The mineral contents of the diets were determined according to requirements of breeds

The mineral contents of DPP were determined according to AOAC [19] by using atomic absorption spectrophotometer (PerKin Elmer 100).

Comment 14:

Why minerals with words and symbols? in the text ok, but not in tables.

 Answer:

The symbols were inserted in tables 2

Comment 15:

Row 106: not clear

Answer:

Rewrite to be

Dimension (42width × 50 length × 40 height cm) and equipped with feeders and automatic nipples.

Comment 16:

Row 122-129: check all very well, many mistakes.

Answer:

The paragraph rewrite to be

Shape index (ShI %), Yolk index (YI %), Unit surface shell weight (USSW mg/cm²), and shell (%) were measured by following equations according to Reddy et al. [22] and Anderson et al. [23]. ShI (%) = [Egg width/Egg height] ×100, YI (%) = (Yolk height/Yolk diameter) ×100, Egg surface area (cm²)= 3.9782 × egg weight0.7056, USSW (mg/cm²) = Egg weight (mg)/Egg surface area (cm²), Shell (%) = [Shell weight (g)/ Egg weight (g)] ×100, Yolk diameter (mm) along the chalazae line was determined with the caliper.

Comment 17:

Row 134: how many hours of fasting?

Answer:

All birds were fasted to 6 hours before blood collection

Comment 18:

Row 155: check it

Answer:

28 laying hens were slaughtered

Comment 19:

Row 158-159: not clear

Answer:

The diameter of the largest 1st, 2nd, 3rd, 4th and 5th follicle and total yolk follicle (TYF) were  measured  by  using  a  digital  calliper  within  ±  0.01  mm

Comment 20:

Row 168: not clear

Answer:

using the GLM procedure of SAS-6.03 [28] according the following linear model was applied:

The sentence rewrite to be using the general linear model procedure of SAS-6.03 [28] as follow:

Comment 21:

Row 176: supplementation on egg weight, ...and the other traits?

Answer:

Supplementation on BW, egg weight, egg mass, egg production rate

Egg production rate in the title of table 3

Comment 22:

Table 4: surface area: how it was measured?

Answer:

It was measured by using the flowing equation: Egg surface area (cm²)= 3.9782 × egg weight0.7056

Comment 23:

USSA or USSW?

Answer:

USSW (mg/cm²) was measured by using the flowing equation: Egg weight (mg)/Egg surface area (cm²)

Comment 24:

Yolk index: units?

Answer:

Yolk index (%)

Comment 25:

All the variables needs a comment on the effect.

Answer:

The sentence (Shape index, shell percentage and yolk index were not affected by DPP supplementation) was inserted in line 202 in corrected word.

Comment 26:

row 228: other variables in the tables.

Answer:

Rewrite to be “Liver, spleen, ovary, oviduct and ovarian follicle weights”

The authors thank the Reviewer for reviewing the paper.